# Self-Esteem Is Independent Factor and Moderator of School-Related Psychosocial Determinants of Life Satisfaction in Adolescents

**DOI:** 10.3390/ijerph19095565

**Published:** 2022-05-03

**Authors:** Zsuzsa Lábiscsák-Erdélyi, Ilona Veres-Balajti, Annamária Somhegyi, Karolina Kósa

**Affiliations:** 1Department of Physiotherapy, Faculty of Public Health, University of Debrecen, 26 Kassai St., 4028 Debrecen, Hungary; balajti.ilona@sph.unideb.hu; 2Schools for Health in Europe Network Foundation, National Center for Spinal Disorders, 1-3 Kiralyhago St., 1126 Budapest, Hungary; annamaria.somhegyi@bhc.hu; 3Department of Behavioral Sciences, University of Debrecen, 22 Moricz Zsigmond St., 4032 Debrecen, Hungary; kosa.karolina@med.unideb.hu

**Keywords:** life satisfaction, psychosocial factors, school setting, adolescents

## Abstract

Our aim was to investigate the impact of the school psychosocial environment, including students’ general attitude towards the school, perception of support from teachers and classmates as well as individual psychosocial factors including self-esteem and loneliness on life satisfaction (LS). Four repeated cross-sectional online questionnaire surveys were carried out between 2011 and 2014, inviting all students in one Hungarian high school. Health status and behaviour were assessed by the Hungarian version of the HBSC questionnaire. Results from the surveys were pooled for analysis (N = 3310 students). Heteroskedastic regression estimating robust variance was used to identify potential determinants of LS. Family wealth perceived to be well-off, self-esteem, and being perceived as a good student were identified to be the most important significant positive contributing factors of LS. Perceived good relations with classmates and teachers and an overall positive attitude to school had smaller but still significant positive effects on LS. Self-esteem was a significant moderator for the effect of perceived difficulty of schoolwork, relation with classmates, and gender. This paper shows that self-esteem is not only an independent factor but also a modifier of some school-related variables on LS. The complex interplay among school-related and individual potential determinants of LS should be taken into account in future research by controlling for their interactions.

## 1. Introduction

Schools create one of the most important psychosocial environments for adolescents in public education. Youngsters spend considerable amounts of time in school, with the aim of learning facilitated by the school environment. However, there are a number of factors giving rise to school-related stress, such as interactions between students and teachers, interactions among students, and expectations related to academic performance. These factors have considerable impact on the health and well-being of students shown by a number of studies in the past two decades, many of them from Scandinavian countries. An early study of Torsheim et al. [1] showed that the sense of coherence and school-related stress interacted with subjective health complaints in Norwegian adolescents. Analysis of the 2014 Finnish cohort of Health Behaviour in School-aged Children (HBSC) gave rise to a number of important papers. Markkanen et al. [2], using a comprehensive set of questions, found in that cohort that positive perceptions of the psychosocial school environment were associated with higher levels of subjective health; Lyyra et al. [3] showed loneliness to be a strong predictor of health complaints. Simonsen et al. [4] found in the same cohort that an empowerment-enabling school environment (capturing support from classmates and teachers, and student participation) along with empowering home environments were associated with better self-rated health. Data of the Lithuanian cohort of HBSC from the same year (2014) provided evidence for high levels of school demand and low social support being strong risk factors for multisite health complaints [5]. School psychosocial environment has been linked to various health and risk behaviours among students in the US [6], Canada [7], Bulgaria [8], Portugal [9], and Sweden [10]. The school environment also makes an impact on students’ mental health, as it was shown in a multi-country analysis of time trends in adolescent mental well-being across five HBSC surveys. While large variations were found among countries, declines in mental well-being between 2002–2018 were partially explained by an increase in schoolwork pressure with no change in life satisfaction [11].

This was an intriguing finding since life satisfaction is an important construct of positive psychology that is sensitive to the entire spectrum of functioning. Life satisfaction is appropriate for reporting positive sentiments, not only psychopathological problems as opposed to classical mental health scales. Life satisfaction in youth is associated with physical health, substance abuse, achieving personal standards, self-efficacy, social relations, becoming a victim in a dating relationship, sexual behaviour, as well as mental disability and eating disorders [12]. Self-esteem is another an important concept of positive psychology which captures thoughts and feelings of an individual about their worth and importance [13]. A lower level of self-esteem is associated with an increased risk of mental health problems in adolescents [14], and is correlated with feelings of loneliness partially mediated by perceived social acceptance [15]. Self-esteem was found to partially mediate the effect of parenting practices on life satisfaction in adolescents [16].

The association between life satisfaction and self-esteem was investigated by researchers in samples independent of HBSC surveys. Moksnes et al. [17] found positive associations between the two variables but no interaction effect of gender and self-esteem or age and self-esteem was found in relation to life satisfaction. Gómez-Baya et al. [18] stated that higher self-esteem predicted more emotion-focused positive rumination, and more dampening predicted lower life satisfaction among adolescents. The results of Freire T. et al. [19] highlight the role of life satisfaction and self-esteem for adolescents to manage, regulate, or minimize their psychological distress and to build higher levels of subjective happiness as a source of positive mental health. Low family income was associated with lower self-esteem and life satisfaction in a representative sample of British 11-year-olds by Bannink et al. [20]. After adjustment for income, young people who perceived their family to be poorer or richer than their peers had lower self-esteem and life satisfaction. 

Hansen E. et al. [21] suggested participation in cultural activities in public health promotion programs as it was positively associated with self-perceived health, life satisfaction and self-esteem in adolescents.

Our aim was to investigate the impact of the school psychosocial environment, including students’ general attitude towards the school, perception of support from teachers and classmates, as well as individual psychosocial factors such as self-esteem and loneliness on life satisfaction in a homogenous body of high school students in repeated cross-sectional surveys. 

## 2. Methods

### 2.1. Study Design

Four repeated cross-sectional surveys were carried out to evaluate an integrated health promoting program implemented in the spring of 2011 in one Hungarian school, as described elsewhere [22]. A baseline survey of pupils was carried out before the start of the programme (baseline, April 2011, N = 934), a second survey took place 6 months later (N = 816), and the third (N = 829) and fourth (N = 731) surveys were implemented 1 and 2 years after the second survey, respectively. 

### 2.2. Variables

The questionnaires in the four surveys were identical, all items had been taken from the Hungarian version [23] of the Health Behaviour of School-aged Children (HBSC) 2010 survey [24]. Demographic data included date of birth, gender (male/female), school grade (9 to 12), type of permanent place of residence (county seat, city, village, farm) which was dichotomized into “city” and “not city”. Socioeconomic status of the pupils’ families was assessed by a single item on perceived family wealth (PFW, “How well off do you think your family is?”) answerable on a five-point Likert scale ranging from “not at all well off” to “very well off”. PFW was dichotomized by collapsing answers of “well or very well” into one category and all other responses to another category (“less than well”). 

Life satisfaction was measured by the 11-step Cantril ladder [25] on which the best possible life had to be marked between 0 (worst) and 10 (best). Self-esteem as the evaluative component of the self-concept of pupils was assessed by the Hungarian version [26] of Rosenberg’s Self-Esteem Scale [13] (RSES). The scale consists of 10 statements answerable on a 4-point scale. A summary score was created for each respondent after reversing answers on the appropriate five items so that higher scores reflected higher self-esteem. The Cronbach alpha of self-esteem was 0.876. 

Perceived loneliness was assessed using one item on global loneliness (“Do you ever feel lonely?”) answerable on a 4-point scale (1: never, 2: yes, sometimes, 3: yes, quite often, 4: yes, very often). Responses were dichotomized by creating the reference category including those who are never lonely and collapsing all other answers to the other category (“lonely”). Academic achievement of the students was self-rated by one item asking how the teacher evaluates their academic performance compared to their classmates, which was answerable on a 4-point scale (very good, good, average, below average). This was also collapsed into a binary variable including those who answered “very good” and “good” in one response category (“good student”), and the other two responses in the other category (“not good student”). 

Altogether, 14 questions were asked about pupils’ attitudes towards their teachers (1–4) and classmates (5–8), opinion on schoolwork (9–13), and emotional connectedness to their school (14), all taken from the Hungarian version of HBSC [23] shown in Table 1. 

Spearman correlation analysis revealed that all 14 items were significantly correlated with life satisfaction (*p* < 0.001), but the pairwise correlation for certain items was low or nonsignificant (Table 2) warranting further investigation. Exploratory factor analysis of the 14 items revealed two factors, one capturing students’ perception of their teachers (items 1–4), the other of their classmates (items 5–8). Items 9–14 had high uniqueness (>0.45) loading on two additional factors. These were investigated separately, revealing an acceptable reliability of the summative rating scale of items 9–11 reflecting the difficulty of schoolwork, while items 12–14 captured a general attitude to school. Both were deemed satisfactory and used separately in predicting life satisfaction.

In sum, altogether four scales were created by summing up responses on the corresponding items of the scales after reversing the answers so that higher points reflected greater satisfaction with teachers (items 1–4 in Table 1, Cronbach alpha = 0.801) and classmates (items 5–8 in Table 1, Cronbach alpha = 0.824); greater burden of schoolwork (items 9–11 in Table 1, Cronbach alpha = 0.784); and more positive attitude to school (items 12–14 in Table 1, Cronbach alpha = 0.731). Differences in scale were accounted for by calculating and graphically depicting standardized coefficients after regression. 

### 2.3. Upper Rows: Spearman’s Rank Correlation Coefficient, Lower Rows: p Value Data Collection

A web-based questionnaire was developed for data collection with a standard Linux server using PHP and MySQL support, described in detail elsewhere [22]. The questionnaire could be completed in 20 min. Access to the questionnaire was pre-organized in a scheduled timepoint for groups of those students in the computer room of the school whose guardians consented to their participation. The test was not available outside of scheduled times. The same questionnaire was used in all four surveys in April 2011 (survey1), September 2011 (survey2), October 2012 (survey3), and October 2013 (survey4) so they were fully comparable. 

### 2.4. Data Analysis

Upon completion of the questionnaire, data were automatically logged in a database from which data were downloaded in a Microsoft Excel file. Records were checked for duplicates, empty records and answers out of the specified ranges. After cleaning, data analysis was carried out in MS 365 Excel and STATA IC 16.1. Continuous variables were compared by t-test, categorical variables were analysed by the chi-square test. Normality testing based on skewness and kurtosis was used to test normality of life satisfaction. Though LS is commonly dichotomized [27,28], but because of problems such as the loss of information related dichotomization [29], we used it as a continuous interval variable. The Breusch–Pagan test was used to check heteroskedasticity [30]. Correlation between variables was tested by Pearson product-moment correlation (if continuous) and Spearman’s rank order correlation (if categorical). One-way analysis of variance and the one-way ANOVA on ranks (Kruskal–Wallis test) were used for describing life satisfaction by categorical variables. Variance of life satisfaction was heteroskedastic by grade and self-esteem; therefore, model selection was performed testing ordinary least squares regression and heteroskedastic linear regression with the Huber/White sandwich estimator of variance to account for outliers. This method of regression was shown previously to provide reliable estimates [31]. A coefficient plot was created according to Jann [32]. The level of significance was set at 0.05.

## 3. Results

The four surveys took place in one high school in the second largest city of the country, producing a large pooled sample (N = 3310). Response rates were 77.67% for the baseline survey, 70.41% for the second, 60.41% for the third, and 64.36% for the fourth survey, calculated from the total number of registered students in each survey year. 

Life satisfaction had skewness −1.51 and kurtosis: 7.16. According to Byrne [33], data can be considered normal if skewness falls between −2 to +2 and kurtosis is between −7 to +7. The nonparametric Kruskal–Wallis test was used to describe life satisfaction by survey year and grade in bivariate analysis. 

A significant 0.33 point decrease was detected in the pooled data from grade 9 to grade 12 in bivariate analysis (grade 9: 8.10 ± 1.47; grade 10: 7.94 ± 1.50; grade 11: 7.86 ± 1.55; grade 12: 7.77 ± 1.62) (*p* < 0.001). There was a negative linear fit between life satisfaction and grade, as shown in Figure 1. Moderately significant difference was found in life satisfaction by survey year in bivariate analysis with no clear time trend (survey 1: 7.83 ± 1.50; survey 2: 7.98 ± 1.57; survey 3: 7.95 ± 1.60; survey 4: 7.90 ± 1.53) (*p* = 0.025).

There was no difference in gender distribution (*p* = 0.607), type of permanent residence (*p* = 0.682), and perceived family wealth (*p* = 0.276) by survey year in bivariate analysis, so these variables are described for the entire sample. Girls comprised the majority in all grades in all survey years, varying between 54.05% and 70.29%. A total of 77.6% of the pupils were city dwellers, 22.4% lived in villages or farms, with no significant gender difference in permanent residence (*p* = 0.071). A total of 29.7% of the students lived in families perceived to be well or very well off, 63.5% perceived their families as average, whereas 6.8% of them lived in families not so well or not at all well off. Gender difference of life satisfaction was not significant in any grade (grade 9: *p* = 0.761; grade 10: *p* = 0.471; grade 11: *p* = 0.846; grade 12: *p* = 0.095).

### 3.1. Model Selection

In order to select independent variables for modelling, Spearman correlation was used to check for correlation between life satisfaction and demographic, individual and school-related psychosocial variables. Survey date was related to neither life satisfaction (*p* = 0.157) nor self-esteem (*p* = 0.730), so it was omitted, and the pooled records were used for modelling. Gender showed no correlation with life satisfaction (*p* = 0.707) but showed significant correlation with self-esteem (*p* < 0.001), so it was kept in the tested models along with other variables significantly associated with life satisfaction: demographic binary variables such as place of residence and perceived family wealth; individual psychosocial variables such as self-esteem, perceived loneliness, and self-rated academic achievement; variables of the school psychosocial environment such as perception of teachers, perception of classmates, difficulty of schoolwork, and general attitude to the school, as well as grade as a proxy of age.

To find the best model for predicting life satisfaction, model selection was performed taking into account the heteroskedasticity of life satisfaction by self-esteem (*p* < 0.001). Each of the five models included all independent variables, and differed by method of regression, estimation of variance (standard error), self-esteem in a supposed linear or nonlinear association with LS, and interactions (Table 3). The best model to estimate life satisfaction based on the smallest Akaike information criterion was Model 5. 

### 3.2. Independent Variables of Life Satisfaction

Model 5 applying heteroskedastic regression and modelling the non-linear association between life satisfaction and self-esteem was used to identify significant determinants of life satisfaction. The Huber/White sandwich estimator of variance was calculated to produce robust standard errors as described in Methods. Interactions between self-esteem and its significant school-related psychosocial covariates such as perception of classmates (*p* < 0.001), difficulty of schoolwork (*p* < 0.001), and gender (*p* < 0.001) were also investigated. Perception of teachers (*p* = 0.150) and general attitude to school (*p* = 0.407) were not significantly correlated with self-esteem so only their main effects on LS were analysed. 

Family wealth perceived as well-off compared to not well-off proved to be a major determinant of life satisfaction, since well-off family wealth was associated with a 0.35 point increase in life satisfaction (*p* < 0.001). Self-esteem was another important determinant of life satisfaction: one point increase in self-esteem was associated with a 0.23 point increase in life satisfaction (*p* < 0.001). Being considered a good student was associated with a 0.17 point increase in LS compared to being a not good student (*p* = 0.003). Favourable perception about classmates (b = 0.11; *p* = 0.047), favourable general attitude to school (b = 0.09; *p* < 0.001), and favourable perception of teachers (b = 0.03; *p* = 0.016) had smaller but significant positive effects on LS. In opposition, perceiving schoolwork to be more difficult decreased life satisfaction (b = −0.17; *p* = 0.004), and being lonely sometimes or often was associated with a large, 0.58 point decrease in life satisfaction compared to those who were never lonely (*p* < 0.001) (Figure 2).

Being a girl (compared to being a boy) (*p* = 0.847), living in a city (compared to a village) (*p* = 0.062), or grade (*p* = 0.153 or greater for all four years) had no significant association with LS. 

In terms of interactions, difficulty of schoolwork on LS was significantly modified by self-esteem so that the negative effect on LS of one point increase in perceived difficulty of schoolwork was largest at the lowest level of self-esteem (at score 10: b = −0.125, *p* = 0.001) compared to higher scores (at score 20: b = −0.084, *p* < 0.001; at score 30: b = −0.042, *p* < 0.001) whereas the difficulty of schoolwork had no impact on life satisfaction at the highest level of self-esteem (b = −0.001, *p* = 0.951) (Figure 3). 

Association of the perception of classmates with LS was also modified by self-esteem. Among those with the lowest score of self-esteem (score of 10), 1 point increase in the positive perception of classmates predicted a significant 0.083 point increase (*p* = 0.035) in life satisfaction, but this effect decreased with increasing self-esteem: at the self-esteem score of 20, 1 point increase in the perception of classmates predicted a 0.052 point increase in LS (*p* = 0.017) which tapered off (at score 30: b = 0.02, *p* = 0.045), disappearing at the highest level of self-esteem (at score 40: b = −0.010, *p* = 0.608) (Figure 4). In other words, the positive effect of favourable perception of classmates on life satisfaction was strongest among those with low self-esteem but tapered off as self-esteem increased. 

Self-esteem also modified the effect of gender on life satisfaction, though the overall effect of gender on LS was not significant in the regression model. Gender had no effect on LS at the lowest scores of self-esteem (at score 10: b = 0.033, *p* = 0.877, at score 20: b = 0.128, *p* = 0.280) but became a significant determinant at higher levels of self-esteem (at score 30: b = 0.223, b < 0.001; at score 40: b = 0.318, *p* = 0.003), so that girls had higher life satisfaction compared to boys when their self-esteem was equal (Figure 5). This graph also shows the quadratic (nonlinear) component of self-esteem on life satisfaction (*p* < 0.001) identified in models 4 and 5 (see Table 3).

## 4. Discussion

Four cross-sectional surveys repeated in one high school were pooled to identify school-related and individual determinants of life satisfaction. Model selection identified heteroskedastic regression with robust variance estimator as the best model to test the effect of demographic, individual, and school-related psychosocial variables on life satisfaction. LS decreased with no gender difference from grade 9 to grade 12 in bivariate analysis, but the effect of grade disappeared from the full model. Of all investigated variables, perceived family wealth and self-esteem were the most important positive while having difficulties with schoolwork and being lonely were major negative determinants of life satisfaction. In addition to a significant main effect, self-esteem was shown to be moderating the effect of difficulties with schoolwork and perception of classmates, mostly at lower levels and tapering off at the highest level. 

The surveys were implemented in an elite high school in Hungary which has been in the top 100 high schools in the country. More than 88% of the students lived in families which they perceived as being in average wealth or better, so the family background of the students could be considered relatively homogenous, minimizing the well-known impact of the socioeconomic conditions of the family on adolescent LS [34]. The surveys were carried out within the framework of a health promoting program details of which are being published elsewhere [22]. Repeated cross-sectional design allowed evaluation of the stability of school-related determinants, possible only in such surveys using identical methods of data collection. However, since the student body slightly changed from one survey to the next as grade 12 students left and grade 9 students entered the school, and there was no repeated measures design, cause–effect relationships could not be determined. Invitation of all pupils of the school to participate decreased the risk of selection bias, and pooling all surveys produced a large sample. Students who did not participate because of refusal, lack of parental consent or absence on survey days might have been different from the rest of the students in terms of life satisfaction, but considering the satisfactory participation rate (varying between 60–78%) and repeated data collections, valid estimates can be made.

One of the limitations of the study is the use of self-assessment scales, such as the one for perceived family wealth. Objective data such as parental education and number of computers in the household were collected to assess the socioeconomic status of the family. We conducted a literature review before data analysis in order to determine which variables should be used to assess the family’s socioeconomic status, and found that perceived family wealth was more frequently used than its objective alternatives. Subsequently, we created a composite indicator from the following objective parameters: type of permanent place of residence (county seat, city, village, farm), parents’ highest educational qualification (higher education diploma, high school graduation, vocational school, completed primary school, less than primary school), and number of computers in the household (none, one, two, more than two) with a total score ranging between 4–18. This composite indicator was correlated with perceived family wealth yielding a correlation coefficient (Spearman’s rho) of 0.361 (*p* < 0.001). Therefore—in line with the findings of the literature search—we used perceived family wealth in all subsequent analysis. Another limitation may be that data collection occurred between 2011 and 2014.

Life satisfaction among teenagers had been investigated by a number of research groups. The KIDSCREEN project funded by the EU and implemented between 2001–2004 developed standardized instruments to help assess children’s quality of life; self-perception measured by one of these instruments was identified as a major predictor of LS among Polish children [35]. Life satisfaction measured by the single-item Cantril ladder was introduced as a mandatory variable in the Health Behaviour in School-aged Children (HBSC) project in 2001/2002 [22]. Some research groups found evidence for the impact on LS of behaviour such as physical activity [28,36], others addressed the effect of socioeconomic status [34] including parental unemployment [37]. Cavallo et al. [38] analysing life satisfaction in 31 countries had shown that LS varied between countries and exhibited different time trends in Western and Eastern European nations. They found LS to decrease with age, which was seen as a non-significant trend in our sample. They also found LS to be lower among girls compared to boys in alignment with other researchers [2,36,39]. Our results did not reveal a significant gender difference in LS; moreover, we found that girls have higher levels of life satisfaction compared to boys in our model if both have equally high levels of self-esteem. 

Social support, meaning the provision of tangible and intangible resources to help recipients cope with stress, has been shown in a meta-analysis to be a significant predictor of well-being in adolescents. The association between social support and well-being was stronger among female adolescents compared to males [40]. The school is an important source of social support for schoolchildren coming from teachers and classmates. A recent study of 2017/18 HBSC surveys from 42 countries investigated the effect of perceived social support on LS, concluding that while support from family had the strongest association with life satisfaction, the second strongest association was shown between support from teachers and classmates and life satisfaction in majority of countries [38]. Similarly, our results showed that perception of teachers and classmates are significant positive determinants of life satisfaction. However, the positive effect of classmates on LS decreased with increasing self-esteem. Support from family was not measured in our study, but we can surmise that the large positive effect of perceived family affluence on life satisfaction was likely mediated by social support within the family [41]. 

Association between life satisfaction and school psychosocial environment was investigated in three cross-sectional HBSC surveys. The Portuguese survey of 2010 classified academic achievement, social competence, and self-regulation as individual assets, and family support, peer support, parental monitoring and school connectedness as social assets. Both these types of assets significantly and independently predicted life satisfaction in adolescents, social assets explaining a greater share of variance in LS than individual assets [42]. Analysis of the Finnish HBSC cohort of 2014 provided evidence that high levels of school engagement, good student relations, and low school strain are associated with higher life satisfaction both in girls and boys [2]. A study of the Swedish HBSC cohort of 2018 reported that lower levels of school demands, and higher levels of support from teachers and classmates had positive association with life satisfaction [43]. Our results are in alignment with these findings, with the important addition that self-esteem is a modifier of the positive effect of perceived relationship with classmates on LS. 

While social support is a positive determinant of life satisfaction among adolescents, schoolwork-related stress may have detrimental effects. Stress related to school performance was associated with depressive symptoms mediated partly by life satisfaction in a sample of rural secondary school students in Norway [44]. Our findings not only showed the negative effect of increasing difficulty of schoolwork on LS but provided evidence for the important effect of self-esteem which modifies the effect of perceived difficulty of schoolwork on life satisfaction.

Self-esteem among adolescents in relation to mental health was investigated in the 2018 HBSC cohort of Scandinavian countries. Structural equation modelling revealed that loneliness was a significant risk factor for both mental wellbeing and self-esteem in all four countries [45]. 

## 5. Conclusions

From the results described above, two conclusions may be derived. We have included a rather wide range of school-related and individual variables in a carefully selected regression model, and to our best knowledge, we are the first to show the effect of the school psychosocial environment with self-esteem as a modifier on life satisfaction. 

In light of our findings, the complex interplay among school-related determinants of life satisfaction should be taken into account in future research by controlling for their interactions. Particular attention should be paid to the differential needs of pupils with high and low self-esteem in terms of school support.

## Figures and Tables

**Figure 1 ijerph-19-05565-f001:**
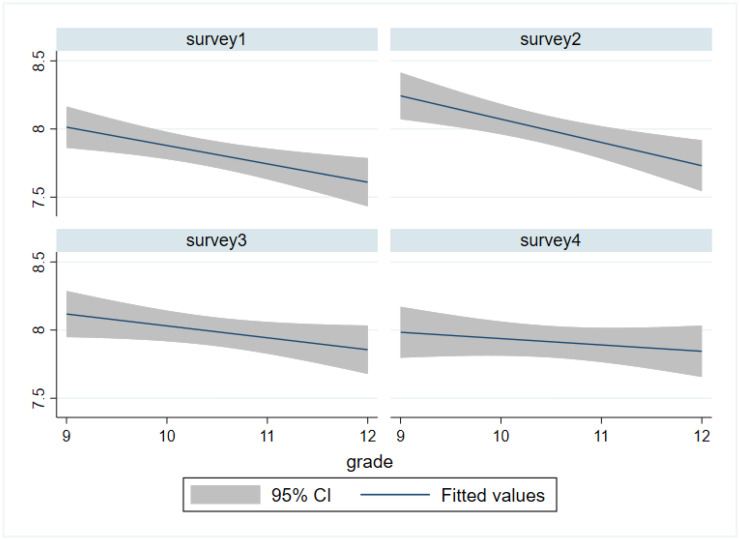
Linear fit with confidence intervals of life satisfaction by grade and survey.

**Figure 2 ijerph-19-05565-f002:**
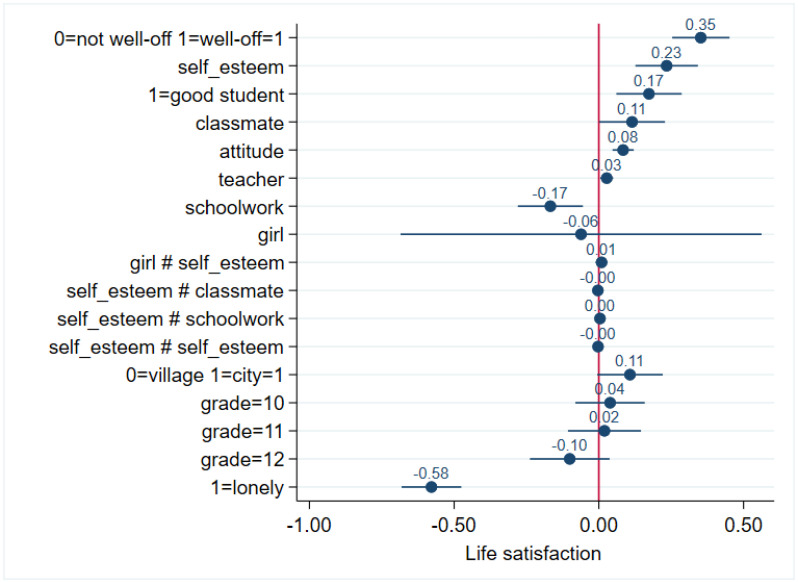
Coefficients with confidence intervals of the independent demographic and school-related psychosocial variables on life satisfaction in adolescents. # means interaction.

**Figure 3 ijerph-19-05565-f003:**
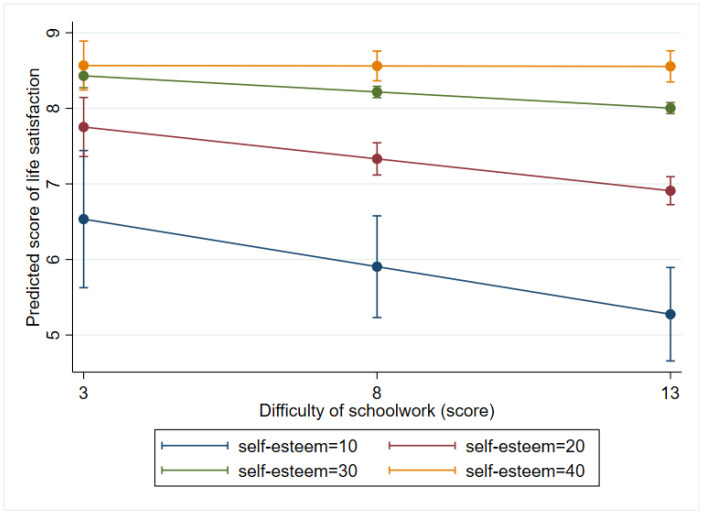
Marginal effects of the difficulty of schoolwork on life satisfaction at different levels of self-esteem.

**Figure 4 ijerph-19-05565-f004:**
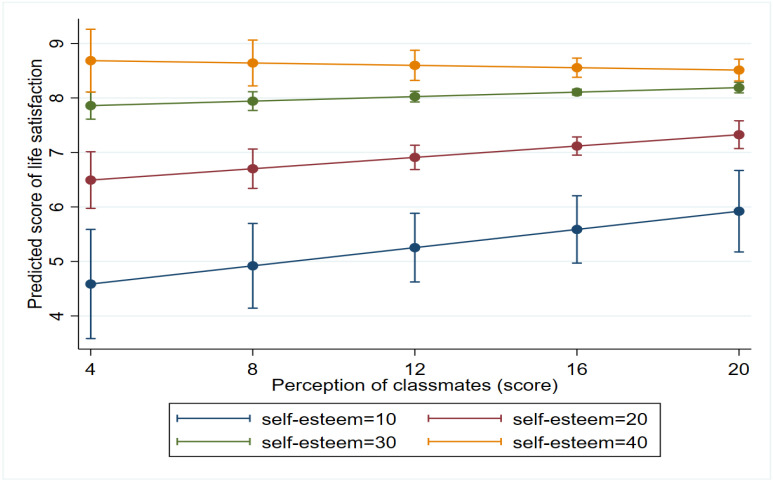
Marginal effects of the perception of classmates on life satisfaction at different levels of self-esteem.

**Figure 5 ijerph-19-05565-f005:**
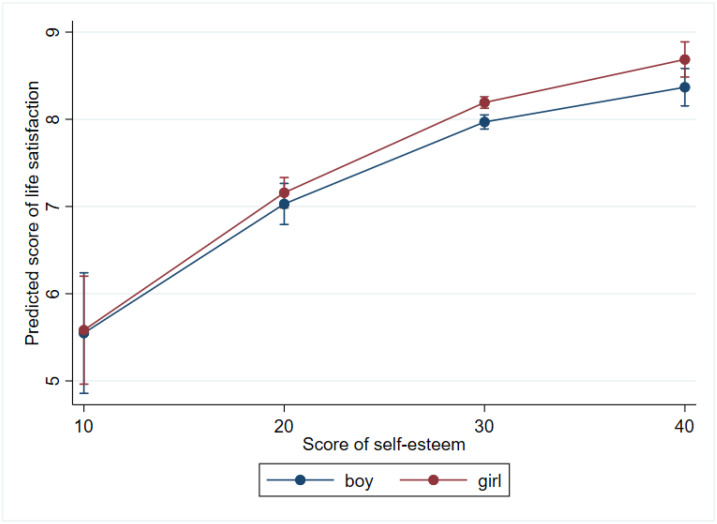
Marginal effects of gender on life satisfaction at different levels of self-esteem.

**Table 1 ijerph-19-05565-t001:** Items related to pupils’ attitudes towards teachers, classmates, schoolwork, and school.

#	Item	Value Range	Responses
1	My teachers encourage me to share my opinion.	1–5	1: strongly agree 5: strongly disagree
2	Our teachers are just with us.	1–5	1: strongly agree 5: strongly disagree
3	If I need special help, my teachers will provide it.	1–5	1: strongly agree 5: strongly disagree
4	My teachers are interested in my personality.	1–5	1: strongly agree 5: strongly disagree
5	The students in my class enjoy being together.	1–5	1: strongly agree 5: strongly disagree
6	Most of the students in my class are kind and helpful.	1–5	1: strongly agree 5: strongly disagree
7	Other students accept me as I am.	1–5	1: strongly agree 5: strongly disagree
8	If someone feels down, there is always someone trying to help.	1–5	1: strongly agree 5: strongly disagree
9	I have too much schoolwork.	1–5	1: strongly agree 5: strongly disagree
10	I think school is difficult.	1–5	1: strongly agree 5: strongly disagree
11	I think school is tiresome.	1–5	1: strongly agree 5: strongly disagree
12	We are learning about interesting stuff in school.	1–5	1: strongly agree 5: strongly disagree
13	I am happy to learn in this school.	1–5	1: strongly agree 5: strongly disagree
14	How do you feel about school at present?	1–4	1: I like it a lot 4: I don’t like it at all

**Table 2 ijerph-19-05565-t002:** Spearman’s rank correlation coefficients with p values for the correlation of school-related items with life satisfaction (LS).

	LS	1	2	3	4	5	6	7	8	9	10	11	12	13	14
LS	1.0000														
1 share opinion	−0.1512	1.0000													
	0.0000														
2 teacher just	−0.1643	0.4604	1.0000												
	0.0000	0.0000													
3 teacher helps	−0.1605	0.4281	0.4877	1.0000											
	0.0000	0.0000	0.0000												
4 personality	−0.1545	0.4908	0.4326	0.4871	1.0000										
	0.0000	0.0000	0.0000	0.0000											
5 enjoy being together	−0.1383	0.2104	0.2209	0.2364	0.2618	1.0000									
	0.0000	0.0000	0.0000	0.0000	0.0000										
6 students kind	−0.1720	0.2246	0.2412	0.2560	0.2609	0.6731	1.0000								
	0.0000	0.0000	0.0000	0.0000	0.0000	0.0000									
7 students accept me	−0.2300	0.1731	0.1903	0.2281	0.2142	0.5139	0.6253	1.0000							
	0.0000	0.0000	0.0000	0.0000	0.0000	0.0000	0.0000								
8 someone helps	−0.1186	0.1768	0.1672	0.2218	0.2069	0.4151	0.4821	0.4746	1.0000						
	0.0000	0.0000	0.0000	0.0000	0.0000	0.0000	0.0000	0.0000							
9 too much schoolwork	0.0917	−0.0699	−0.1349	−0.0758	−0.0716	0.0173	−0.0044	0.0056	0.0199	1.0000					
	0.0000	0.0001	0.0000	0.0000	0.0000	0.3235	0.8029	0.7471	0.2554						
10 school difficult	0.1239	−0.0503	−0.1171	−0.0813	−0.0479	0.0444	0.0116	0.0115	0.0352	0.5873	1.0000				
	0.0000	0.0040	0.0000	0.0000	0.0061	0.0111	0.5085	0.5111	0.0443	0.0000					
11 school tiresome	0.1656	−0.1710	−0.1944	−0.1351	−0.1639	−0.0701	−0.0821	−0.0355	−0.0302	0.5309	0.5289	1.0000			
	0.0000	0.0000	0.0000	0.0000	0.0000	0.0001	0.0000	0.0426	0.0845	0.0000	0.0000				
12 school interesting	−0.1243	0.3487	0.3565	0.3783	0.3197	0.2024	0.2163	0.1665	0.1998	−0.0824	−0.0396	−0.1845	1.0000		
	0.0000	0.0000	0.0000	0.0000	0.0000	0.0000	0.0000	0.0000	0.0000	0.0000	0.0234	0.0000			
13 happy in school	−0.2227	0.3113	0.3414	0.3579	0.3002	0.3066	0.3179	0.3304	0.2910	−0.0666	−0.0832	−0.1794	0.3849	1.0000	
	0.0000	0.0000	0.0000	0.0000	0.0000	0.0000	0.0000	0.0000	0.0000	0.0001	0.0000	0.0000	0.0000		
14 feeling about school	−0.2468	0.2805	0.2985	0.2985	0.2664	0.2524	0.2799	0.2607	0.2205	−0.1202	−0.1636	−0.2612	0.3239	0.6131	1.0000
	0.0000	0.0000	0.0000	0.0000	0.0000	0.0000	0.0000	0.0000	0.0000	0.0000	0.0000	0.0000	0.0000	0.0000	

**Table 3 ijerph-19-05565-t003:** Specifics of regression models for predicting life satisfaction.

Model	1	2	3	4	5
Outcome	life satisfaction				
Regression	OLS	OLS	HLS	HLS	HLS
Estimation of variance		robust	robust	robust	robust
Hetero-skedasticity			self-esteem (*p* < 0.001)	self-esteem (*p* < 0.001)	self-esteem (*p* < 0.001)
Nonlinearity				self-esteem^2^	self-esteem^2^
Interaction with self-esteem					classmates, schoolwork, gender
N	3003	3003	3003	3003	3003
p (model)	<0.001	<0.001	<0.001	<0.001	<0.001
df	14	14	16	17	20
AIC	10,045.72	10,045.72	9877.252	9849.1	9841.314

OLS: ordinary least squares, HLS: heteroskedastic linear regression.

## Data Availability

Data are available from the authors at reasonable written request after authorization by the Data Protection Office of the University of Debrecen, Hungary.

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
