# Peer review of "Self-Esteem Is Independent Factor and Moderator of School-Related Psychosocial Determinants of Life Satisfaction in Adolescents"

_ijerph, 2022, doi:10.3390/ijerph19095565_

Round 1

Reviewer 1 Report

The study entitled: "Self-Esteem is an Independent Determinant and Moderator of School-Related Psychosocial Determinants of Life Satisfaction in Adolescents" investigates the impact of various school-related psychosocial variables on life satisfaction in secondary school students, delving into the role of self-esteem as a moderator.

There are several positive aspects to this study.

The research variables considered are interesting because they focus on the psychosocial aspects of the school environment and not just the objective ones.

The study is well structured, clear, and enjoyable to read.

Although the relationships between the variables examined have already been highlighted in previous studies, some of them have been explored in greater depth, emphasizing the importance of self-esteem in moderating the effect between different variables.

However, there are areas for improvement.

In the Introduction:

- it would be good to point out from the outset that the relationship between the variables considered and life satisfaction has already been extensively addressed in numerous researches conducted in different countries around the world (deepen the literature search).

- since previous studies using questionnaires other than the HBSC have already explored many of the relationships between the variables investigated, highlight this aspect before describing specific studies conducted in Europe using the HBSC;

- for future research, in the school environment, rather than self-esteem, it would be interesting to investigate the role of self-efficacy, which is more task-specific and more sensitive to the influences of the school environment, as previous research has done.

Method

The sample is very large and this is certainly positive.

It would be interesting to further research also with ordinary students and not only "rich" or affluent ones, to verify the real relationship between the variables.

Several scales use a self-assessment that could distort the real data (for example, the perceived wealth scale). It would have been better to use an objective income scale, or a scale designed to assess household cultural capital.

Where self-assessment scales have been used, especially if they feature only one item, indicate this among the research limitations in the discussion.

Ethical issues should be explored further. Has the research been approved by an ethics committee?

Discussion

Interpret results not just in terms of cause-and-effect. The terms "affects" and "determinant" should also be revised in the abstract and perhaps in the title.

Minor revisions

Line 34 Indicate the bibliographic reference immediately after the authors' names. For example, Torsheim [1]... Also in lines 41, 42 and following.

Author Response

Please find our answers in the attached cover letter.

Reviewer 2 Report

Thank you for your research on the associations between adolescent life satisfaction and psychosocial determinants over time. As highlighted in the paper, self-esteem plays a key role in explaining the pattern of associations. Below, I have shared several comments for the authors' consideration:

  1. Review of literature and development of study rationale. I am curious about the theoretical grounding of this work. It would be helpful to establish how this work is based upon theory so as to provide the reader with a better understanding of the planned comparisons.
  2. On page 3, starting on line 117, it looks like there are some words missing or perhaps a typo as the meaning of the sentence is not clear. "Altogether 14 questions were asked about pupils’ attitudes towards their teachers (1-4) and classmates (5-8), opinion on schoolwork (9-13), and emotional connectedness to their school (14), all taken from the Hungarian version of HBSC 18Error! Bookmark not defined. shown in Table 1."
  3. Discussion. Much of the discussion seems like a summary of the results and comparisons to previous research. The comparisons are fine, but as a reader, I was asking myself "so what?" It would be useful for the discussion to help the reader understand the meaning and value (and perhaps usefulness) of the current results in comparison to previous research. 
  4. Implications. In the section where the authors comment on the uniqueness of the project, it would be helpful to explain why these uniquenesses matter. 

Round 2

Reviewer 1 Report

The authors have scrupulously followed the referee's suggestions and the article is definitely improved over the points made in the previous version. 

Author Response

Thank you for all your comments and suggestions.